# Modulation of the Visual to Auditory Human Inhibitory Brain Network: An EEG Dipole Source Localization Study

**DOI:** 10.3390/brainsci9090216

**Published:** 2019-08-27

**Authors:** Rupesh Kumar Chikara, Li-Wei Ko

**Affiliations:** 1Department of Biological Science and Technology, College of Biological Science and Technology, National Chiao Tung University, Hsinchu 300, Taiwan; 2Center for Intelligent Drug Systems and Smart Bio-devices (IDS2B), National Chiao Tung University, Hsinchu 300, Taiwan; 3Institute of Bioinformatics and Systems Biology, National Chiao Tung University, Hsinchu 300, Taiwan; 4Swartz Center for Computational Neuroscience, University of California San Diego, San Diego, CA 92093, USA

**Keywords:** electroencephalography, independent component analysis, dipole analysis, EEG-coherence, brain connectivity, right cingulate gyrus, response inhibition

## Abstract

Auditory alarms are used to direct people’s attention to critical events in complicated environments. The capacity for identifying the auditory alarms in order to take the right action in our daily life is critical. In this work, we investigate how auditory alarms affect the neural networks of human inhibition. We used a famous stop-signal or go/no-go task to measure the effect of visual stimuli and auditory alarms on the human brain. In this experiment, go-trials used visual stimulation, via a square or circle symbol, and stop trials used auditory stimulation, via an auditory alarm. Electroencephalography (EEG) signals from twelve subjects were acquired and analyzed using an advanced EEG dipole source localization method via independent component analysis (ICA) and EEG-coherence analysis. Behaviorally, the visual stimulus elicited a significantly higher accuracy rate (96.35%) than the auditory stimulus (57.07%) during inhibitory control. EEG theta and beta band power increases in the right middle frontal gyrus (rMFG) were associated with human inhibitory control. In addition, delta, theta, alpha, and beta band increases in the right cingulate gyrus (rCG) and delta band increases in both right superior temporal gyrus (rSTG) and left superior temporal gyrus (lSTG) were associated with the network changes induced by auditory alarms. We further observed that theta-alpha and beta bands between lSTG-rMFG and lSTG-rSTG pathways had higher connectivity magnitudes in the brain network when performing the visual tasks changed to receiving the auditory alarms. These findings could be useful for further understanding the human brain in realistic environments.

## 1. Introduction

In daily activities, people generally receive visual stimuli in their surroundings, such as in driving, typing, and sports, although in some emergency situations, people can also receive auditory alarms, like an ambulance siren, a fire truck siren, or a gunshot. In these real situations, people must make a decision to control their response. Therefore, our study investigates how auditory alarms affect the neural networks of human inhibitory control in real situations. The present work used a stop signal task with visual stimuli and auditory alarms. Logan [1] has identified auditory alarms’ importance in real-environmental conditions. For example, the importance of auditory alarms in the executive function of human inhibitory control in the case of an unfocused pedestrian, who is about to leave the footpath in traffic, and a driver at full speed coming at a busy intersection [2]. Moreover, in clinical research, the inability to exercise inhibitory control can interfere with the performance of behavioral goals and contribute to psychological disorders, like attention deficit hyperactivity disorder or obsessive-compulsive disorder [3,4]. In addition, most previous studies on human inhibitory control have evaluated the performance of subjects in visual stimulus experiments with very few studies that have investigated the performance of subjects in auditory alarms [5]. Therefore, in this study, we used auditory alarms with both left-hand response (LHR) and right-hand response (RHR) inhibitions, which increased the complexity of the experimental design. Accordingly, both groups exhibit distinctive EEG patterns; for example, left-handed subjects exhibit less hemispheric asymmetry than right-handed participants in performing complex motor tasks [6,7,8]. Recent studies have examined the only importance of the visual stimulation and they found that the neural activities increased alpha (8–12 Hz) and theta (4–7 Hz) band powers in human brains after the visual stimuli onset [9,10].

The processing of auditory alarm information significantly influences visual stimuli in daily activity. The ability to determine accurately the brain dynamics and neural network pathways of a sound source (i.e., auditory alarms) is important [11]. Auditory information has to be processed in the temporal cortex of the central nervous system, specifically in the primary auditory cortex of the brain [12]. Vision enables an opportunity for the brain to perceive and respond to changes in the human body. The visual information is processed in the occipital lobe of the brain, particularly in the primary visual cortex the brain [13]. In real-environments, the human brain receives several inputs concurrently from its various sensory systems. To combine multisensory information efficiently, the brain must determine whether signals from different channels are associated with a common perceptual event, or give rise to different perceptual events, so whether they must be processed separately. The EEG-coherence of cortical oscillations has been proposed as a potential corollary of multisensory processing and integration [14,15]. In this work, an auditory stop-signal task was designed to measure the neural mechanism of the human brain during visual stimuli and auditory alarms [16]. Additionally, in this study, we utilized an advanced EEG dipole source localization method via independent component analysis (ICA). The ICA process was performed in EEG-Lab and Matlab [17]. After the ICA process, component clustering was analyzed using DIPFIT2 routines, a plug-in EEGLAB, to find the 3D location of an equivalent dipole source location in human brain [17,18,19]. DIPFIT2 function measured the dipole source localization by fitting an equivalent current dipole model using a non-linear optimization technique and using a 4-shell spherical model [17,18,19]. We removed the noisy components from across subjects by visual inspection; after that we used the K-means clustering (K = 5) algorithms in EEG-Lab to identify the best cluster of components from across the subjects [17]. Finally, we obtained five clusters as regions of interest (ROI) in this study.

The main focus of time-frequency analyses is in the neural oscillation activity by determining the time and frequency decomposition of EEG signals. When EEG signals are disturbed through stimulus events, like auditory alarms and visual stimuli, the delivery of the EEG phase becomes phase locked to that event [20]. Neural oscillations of brain activity can be measured by calculating phase associations in all EEG signals [21]. Event-related spectral perturbation (ERSP) is a method used to measure EEG signal activities in various regions of the brain under visual stimuli and auditory alarms with human inhibitory control. It reflects the temporal and spatial resolutions within the EEG signal and elucidates the extent to which the underlying event-related synchronization (ERS) occurs. Therefore, ERSP provides a direct measure of cortical synchrony that is not available from the aggregate evoked response waveform [20]. However, previous studies have reported that human brain perception influences the power spectrum of an ongoing event. The spectral power of an EEG signal that is evoked by external stimulation affects the perceptual performance. Such power spectral modulations of the neural oscillations have been identified in response to visual and auditory stimuli, and have been shown to contribute significantly to the ERSP analysis [20]. Neural oscillations in response to visual stimuli have been identified in the auditory cortex of the brain [22]. In addition, previous research has shown that a visual input restores the role of neural oscillations in the auditory cortex of the human brain [23].

EEG coherence analysis is a mathematical method that can be used to determine whether two or more brain regions exhibit similar neuronal oscillatory activities. Since the 1960s, EEG coherence has typically been evaluated as the similarity of the frequency band across EEG signals. Recently, the coherence has been imaged in the brain, to measure how the neural networks change in several different neurological disorders [24]. The EEG coherence analysis reveals the phase stability between two different time series. The Fourier transform shows a direct relationship between the time and frequency domains and characterizes the alteration of time as a phase difference or phase angle. If the phase angle is constant in time, indicating the phase lock, then the coherence is equal to one, whereas if the time differences between the two-time series differ from one moment to another, then the coherence is equal to zero. EEG coherence is frequently used to analyze the “connection” and as a measure of the functional link between two brain regions [25]. EEG coherence analysis is a sensitive method for elucidating features of an inhibitory brain network. The difference between EEG oscillations is frequently used to calculate “directed coherence,” which is the degree of directionality of a flow of information between two brain regions [26]. Differences in EEG oscillations have also been used to evaluate conduction velocities and synaptic integration times as the inter-electrode distance in different directions increases [27,28].

The functional roles of the left-superior temporal gyrus (lSTG) and the right-superior temporal gyrus (rSTG) are involved with the perception of emotions [29,30]. In addition, the STG is an important structure involved in auditory stimulation, as well as language—particularly among those with poor language skills. It has been revealed that STG is an essential structure in the pathway involving the prefrontal cortex and the amygdala cortex of the brain, which was involved in the development of social cognition [29,30]. Previous research that showed with the use of neuroimaging has found that patients with schizophrenia have structural abnormalities in their superior temporal gyrus (STG) [29,30]. An fMRI analysis study has shown a link between vision-based problem solving and activity in the right anterior superior-temporal gyrus [30]. Moreover, the functional role of right-cingulate gyrus (rCG) has importance in executive function and cognitive control, which involves a set of cognitive processes, including inhibitory control, attention control, and motor control. The role of the rCG is to generate neural impulses that control the execution of movement. The rCG plays a role in the distribution of cognitive resources to synchronized auditory and visual information [31]. Accordingly, clinical research has revealed that the rCG has an essential role in neurological disorders, such as schizophrenia and depression [32,33]. The right-middle frontal gyrus (rMFG) was associated at the stage where response inhibition and sustained attention were supposed to happen. Additionally, the rMFG has been shown to be a crucial area for maintaining visual attention. Therefore, the rMFG has been considered as an important brain region for maintaining attention instead of stopping the action [34,35]. Additionally, the right-parietal lobe (rPL) has been linked to the perception of emotions in facial recognition. However, the rPL is generally considered related to the visual stimuli that are less precisely related to human inhibition function [34]. The rPL plays a functional role in the integration of sensory information from various regions of the human brain. It also plays a role in the processing of information related to the sense of touch [36]. The rPL is involved with visuospatial processing in the human brain [37]. The rPL receives somatosensory and visual information through motor signals and controls the movement of the arm, hand, and eyes [38]. Moreover, changes in brain activity and its associated neural network were investigated under human inhibitory control in five brain regions—(1) the left-superior temporal gyrus (lSTG), (2) right-superior temporal gyrus (rSTG), (3) right-cingulate gyrus (rCG), (4) right-middle frontal gyrus (rMFG), and (5) right-parietal lobe (rPL). We utilized their components and dipole sources’ location to measure the EEG activities during visual stimuli and auditory alarms under human inhibitory control. For the first time our work investigates the effect of auditory alarms on visual stimulation in the human brain. This work revealed how neural networks’ pathways change from visual stimulation to auditory alarms.

## 2. Materials and Methods

### 2.1. Participants

Twelve male university students joined in on the auditory stop signal task. All were between 25 and 30 years old (mean ± SD age: 27.66 ± 1.54 years). All subjects joined the stop signal task for human inhibitory control of both left-hand and right-hand responses. In this study, left-hand and right-hand response inhibitions were used to increase the complexity of the experimental design similar to a real environmental setting; the tendency to use either hand during inhibition is more natural. All participants were right-handed and had no hearing or visual impairment. All participants gave their written informed consent in accordance with the laws of the country and the Research Ethics Committee of National Taiwan University, Taipei, Taiwan. This study was carried out in accordance with the recommendations of the Institutional Review Board (IRB) of the National Taiwan University, Taipei, Taiwan. The study was approved by the Research Ethics Committee of the National Taiwan University, Taipei, Taiwan.

### 2.2. Experimental Design

In this study, all subjects performed a stop-signal task or go/no-go task [1,16]. When the stop trial completes before the go-trial, the subject’s response is inhibited, and when the go-trial completes before the stop trial, the response is permitted. In this experimental paradigm, all the participants carried out a primary task, which in this study was a task of identifying shapes, that required participants to differentiate between a square and a circle (visual stimuli), as presented in (Figure 1). Furthermore, an auditory alarm (beep sound) was used as a stop signal and subjects were instructed to stop their left and right-hand responses. As the delay between the primary task and the stop signal delay (SSD) increases, the probability of response to the stop signal increases [16,39]. This experimental presentation started when participants pressed the ‘Enter’ key on the keyboard. The response keys were ‘Z’ for the square stimulus and ‘/’ for the circle stimulus. The trial was terminated when the ‘ESC’ key was pressed. In the go-trials, participants respond to the shape of square and circle as a visual go stimulus (a “square” requires a left-hand response (LHR) and a “circle” requires a right-hand response (RHR)). In the stop-trials, a beep sound (auditory alarm) was used as a stop signal; we instructed the participants to inhibit their hand response when they heard the beep.

In the go-trials (75% of all trials), only the visual stimuli were used and the participants were instructed to respond to the visual stimuli as quickly and correctly as possible. The size of the visual stimulus was 2.5 × 2.5 cm (height × width, 1 × 1 in). In the stop trials (25% of all trials), the participants were instructed to stop their responses based on the presentation of the auditory alarms. The fixation signal, the square and circular shaped visual stimuli, were shown in white on a black background in the center of a computer display. Each go-trial was started with the display of a white central fixation cross signal for a randomized duration (0.5 s to 6.5 s), followed by the display of a square or circular symbol for 1000 ms. The inter-stimulus interval (ISI) was 1000 ms and it was independent of the reaction time (RT). The visual stimuli remained on the screen until the subjects responded, or until they had passed 1000 ms (maximum RT). In the stop trials, all participants heard an auditory alarm tone binaurally through headphones. This short, (750 Hz, 100-ms-long) beep sound was presented as a stop-signal that indicated to participants that they should inhibit their left or right-hand response in the primary task, regardless of the symbol displayed. In stop-signal trials, a stop signal was presented after a variable stop signal delay (SSD). The SSD was initially set at 250 ms and it was adjusted continuously by the staircase tracking procedure. According to the staircase tracking method, when the inhibition was successful, SSD increased by 50 ms, and when the inhibition was unsuccessful or failed, the SSD decreased by 50 ms. This method increased the complexity of the stop-signal task. The staircase procedure was intended to converge on an SSD that caused the subject to successfully inhibit 50% of the stop trials. This allows for the calculation of the stop signal reaction time (SSRT) by subtracting the critical SSD latency from the RT [16,34,35]. In this study, 60 stop trials and 180 go trails were executed for each subject. The behavioral performance feedback parameters, such as RT, SSD, SSRT, hit percentage, and miss percentage, were investigated according to the presentation of visual stimuli and auditory alarms.

### 2.3. Acquisition and Pre-Processing of EEG Signals

The EEG signals were acquired from healthy subjects using a Scan NuAmps Express system (Compumedics USA Inc., Charlotte, NC). The EEG cap of 32 channels was used with the international 10–20 system for electrode positioning. First, the EEG artifacts were removed manually, after which we performed independent component analysis (ICA) in EEGLAB toolbox [17,18,19]. During the artifact removal process, we observed that about 10% of the epochs were bad in the raw EEG signal. Therefore, the sample size of the EEG data set was reduced 10% by eliminating various artifacts, such as muscle and blinking artifacts [17,18,19]. The EEG data set was down-sampled from 1000 to 500 Hz. To eliminate linear trends, we performed an infinite impulse response (IIR) Butterworth filter using Matlab functions, *filtfilt* and *butter.* The setting of the IIR Butterworth filter was fixed at a high band-pass filter cutoff frequency of 1 Hz, and a low band pass filter cutoff frequency of 50 Hz, after the down-sampling. The EEG signals were preprocessed using MATLAB R2012b (The MathWorks Inc., Natick, MA, USAA) and EEGLAB toolbox (10.2.2.4bVersion, UC San Diego, Swartz Center for Computational Neuroscience (SCCN), La Jolla, CA, USA) [17]. The ICA method was used to eliminate various artifacts, including eye movement and blinking artifacts, and noise from the indoor power line. The decomposition of ICA is a preferred computational method for the separation of blind sources in the processing of EEG signals [18,19]. In this study, we used *runica* algorithms for Infomax ICA decomposition [40]. To identify several types of artifacts, we inspected the scalp map, power spectrum, and dipole source location of each independent component. We executed artifact identification of the EEG signals with scalp map, power spectrum, and dipole location by visual inspection. Based on these criteria we separated good and bad components, back-projecting the retained components to clean the EEG signal. After that, we used the clean EEG signals to perform the ERSP analysis, using functions of the EEGLAB toolbox (10.2.2.4bVersion) [17]. Following the ICA, the EEG dataset was used for ERSP and coherence analysis of visual and auditory stimuli. Each epoch was extracted from –500 to 0 ms as a baseline and from 1 to 1300 ms for the stop and go-trials. The EEG data from the successful-go (SG) trials, successful-stop (SS) trials, and failed- stop (FS) trials were used in the analysis of human inhibitory control. To study brain dynamics after visual stimuli and auditory alarms, each trial was independently transformed into the time and frequency domain through event related spectral perturbation (ERSP) analysis [17]. In this study, we investigated ERSP based on independent component and dipole analysis.

### 2.4. Behavioral Analysis

Behavioral analysis was performed using parameters, such as SSD (i.e., visual reaction time), SSRT (i.e., auditory reaction time) in stop trials, and Go-RT (i.e., visual reaction time) in go-trials. Since the response inhibition reaction time itself cannot be measured directly, the SSRT was calculated by subtracting SSD from the RT. The efficiency of the response inhibition for individual participation was determined. In go-trials, the success rate (hit%) for each participant was acquired from the number of successful responses in the go-trials. In stop trials, the success rate (hit%) for each participant was obtained from the number of successful response inhibitions in stop trials. In go-trials, the miss rate (miss%) was defined as the number of missed responses (excluding the incorrect answers) relative to the total number of go-trials. In stop trials, the miss rate (miss%) was defined as the number of failed response inhibitions relative to the total number of stop trials [16,17].

### 2.5. Independent Component and Dipole Clusters Used as Regions of Interest (ROI)

In our study, independent component clusters were used as regions of interest (ROI). The investigation of independent component analysis (ICA) was performed using EEG-Lab [17]. After ICA processing, component clustering was analyzed using DIPFIT2 routines, an EEGLAB plug-in, to find the 3D location of an equivalent dipole within the brain [17,18,19]. DIPFIT2 performs source localization by fitting an equivalent current dipole model using a non-linear optimization technique and a 4-shell spherical model [17,18,19]. We removed the noisy components and dipoles from across subjects. After that we performed the K-means clustering (K = 5) and dipole-coordinate fitting methods to identify the best cluster of components and diploes across subjects [17]. For the analysis of IC’s clusters, after the decomposition of ICA, we stored the ICA weight matrix (EEG.icaweight) of each IC. The EEG.icaweight matrix of each IC was used for the K-means clustering analysis. For the analysis of dipole clusters, we utilized each diploe’s X, Y, Z coordinates. The K-means clustering analysis was processed using MATLAB R2012b function kmeans with EEGLAB toolbox (10.2.2.4bVersion) [17,40]. Among components and dipoles from all subjects, those with similar scalp maps, dipole locations, and power spectra were clustered. We found five independent component and dipoles clusters, including the left-superior temporal gyrus (lSTG), right-superior temporal gyrus (rSTG), right-cingulate gyrus (rCG), right-middle frontal gyrus (rMFG), and right-parietal Lobe (rPL). We used those five clusters as regions of interest (ROI) in this study. Those five brain regions were selected owing to variances among their visual and auditory modalities, as shown by ERSP analysis.

### 2.6. Analysis of Brain Connectivity under Human Inhibitory Control

In previous studies, EEG-coherence analysis has been recognized as having great potential in studying the human brain’s neural network [24]. EEG signals coherence analysis was conducted herein to quantitatively compute the linear dependency between two EEG signals. The coherence computed between two EEG signals that were acquired at different brain regions of the scalp map reveals a functional relationship between the underlying brain networks. The coherence values, such as, ‘1′ and ‘0′, represent the mutually dependent and unrelated EEG signals, respectively. Coherence was acquired statistically by cross spectral analysis. The methodical aspects of such an EEG-coherence study have been comprehensively evaluated previously [24,41]. The source information flow toolbox (SIFT) in EEGLAB was used to obtain the optimal multivariate autoregressive model [42]. A brain connectivity model was then developed based on the EEG coherence analysis of five dipole clusters under human inhibitory control. We utilized five dipole clusters and their Montreal Neurological Institute (MNI) space coordinates’ X, Y, and Z values to compute the volume of each dipole cube using the open access software “Talairach Client -Version 2.4.3” (This software was developed by Jack Lancaster and Peter Fox at the Research Imaging Institute of the University of Texas Health Science Center, San Antonio, TX, USA), as shown in Table 1. A brain connectivity model was then developed based on the EEG coherence analysis of five dipole clusters (i.e., five brain regions), using combined visual stimuli and auditory alarms during human inhibitory control. These five brain regions were selected owing to variances among their visual and auditory modalities; such inhibitory-control-related neural signals generally arrive from the frontal cortex and the presupplementary motor area [34].

### 2.7. Statistical Analysis

In the behavioral analysis, the ANOVA: Single Factor test was utilized to match the response time between three groups—RT, SSD, and stop signal reaction time (SSRT)—in LHR and RHR inhibitions. In addition, an ANOVA was performed between four groups, the hit% versus miss% in visual stimuli and hit% versus miss% in auditory stimuli, for both LHR and RHR inhibitions. Post hoc comparisons were carried out with Fisher’s least significant difference test. In the EEG analysis, the post-stimulus effects in the brain dynamics from the stop and go-trials were studied by transforming the EEG data after each epoch into the time and frequency domain using the ERSP routine [17]. The statistically significant differences between visual stimuli and auditory alarms in the time-frequency domain were evaluated using the bootstrap method [17,43], with the significance threshold at *p* < 0.05. The mean ERSP value was computed per cluster at each time-frequency domain. For the power spectral analysis, we used the spectopo function in EEGLAB with Matlab [17,20]. We set the parameters for EEG power spectral analysis as follows: The EEG signal sampling frequency, Fs, to 1000 Hz, the length of the EEG signals from −500 to 0 ms as a baseline, and from 1 to 1300 ms as the go-trial and stop trial range. A pairwise *t*-test was used to compare the visual and auditory affects in delta (1–4 Hz), theta (4–7 Hz), alpha (8–12 Hz), and beta (13–30 Hz) spectral bands during response inhibition. Power spectral analysis shows the power spectrum of a time series and describes the power distribution in frequency components, so that we can observe the increase and decrease of brain activity elicited by visual stimuli and auditory alarms. The coherence was acquired statistically by cross spectral analysis in Matlab. The methodical aspects of the EEG-coherence study have been comprehensively evaluated elsewhere [24,41]. The source information flow toolbox (SIFT) in EEGLAB was used to obtain the optimal multivariate autoregressive model [42]. EEG-coherence analysis was used to develop a visual-auditory cross model neural network, as it allows for the computation of connections between different brain regions during human inhibitions [24].

## 3. Results

### 3.1. Behavioral Results

In Figure 2 the different behavioral response times, RT, SSD, and SSRT, for the different tasks are shown; asterisks indicate significance differences (*** *p* < 0.001) between conditions calculated using a single factor ANOVA. The ANOVA test shows a significant difference in LHR between RT (580.48 ± 64.61 ms), SSD (338.92 ± 87.83 ms), and SSRT (241.55 ± 66.31 ms) (*F* (2, 33) = 61.69, *p* < 0.001; Figure 2A). In addition, the ANOVA test shows a significant difference in RHR between RT (574.96 ± 63.23 ms), SSD (356.31 ± 103.34 ms), and SSRT (218.64 ± 85.76 ms) (*F* (2, 33) = 48.35, *p* < 0.001; Figure 2B). Post hoc comparison shows that the RT was observed to be significantly higher than SSD and SSRT, in both LHR and RHR inhibitions.

In Figure 2C,D the ANOVA test shows significant differences in multiple group comparisons, between hit% (i.e., accuracy rate) and miss% (i.e., inaccuracy rate) in visual and auditory stimuli, during LHR and RHR inhibitions. The ANOVA revealed the significant difference in LHR between hits (96.35%) and misses (3.64%) during visual stimuli, and in auditory stimuli between hits (57.07%) and misses (42.92%) (*F* (3, 44) = 82.07, *p* < 0.001; Figure 2C). Moreover, the ANOVA test showed the significant difference in RHR between hits (95.20%) and misses (4.79%) during visual stimulation, and with auditory stimulation, between hits (56.83%) and misses (43.16%) (*F* (3, 44) = 79.25, *p* < 0.001; Figure 2D). The post hoc comparison revealed that the visual stimulus elicited a significantly higher accuracy rate (hit; 96.35 %) than the accuracy rate (hit; 57.07 %) of the auditory stimulus, in LHR and RHR inhibitions.

### 3.2. EEG Results

#### 3.2.1. EEG-Scalp Maps and Dipole Source Locations

In this study, the independent components and dipoles extracted from all subjects with similar scalp maps and dipole source locations were clustered into the same group, as presented in Figure 3. The five dipole clusters were identified on the basis of similar scalp map, power spectra, and diploe source locations across all subjects. We found major changes in EEG activities in the lSTG, rSTG, rCG, rMFG, and rPL of the brain during visual stimuli and auditory alarms, as shown in Figure 3A–E. The dipole source locations and directions of the five clusters of interest were plotted together in Figure 3F. All five clusters and the MNI coordinates of their source distributions during visual and auditory alarms while under inhibitory control are shown in Table 1. These five regions of the brain were used to investigate the effect of visual stimuli and auditory alarms by ERSP, as described below.

#### 3.2.2. Event Related Spectral Perturbation (ERSP) Analysis

Figure 4, Figure 5, Figure 6, Figure 7 and Figure 8 show the average ERSPs of all subjects; non-significant ERSP values are shown in green and significant ERSP values (*p* < 0.05) are shown in yellow and red colors. The ERSP plots were obtained for successful go (SG) (i.e., effect of visual stimuli), successful stop (SS) (i.e., effect of auditory alarms) trials during LHR and RHR inhibitions. The neural mechanisms of the auditory alarms were explored by contrasting the combined audio-visual (AV) and visual (V) stimuli in the ERSP plots under (AV-V) or (SS-SG) conditions. The ERSP maps of cortical distributions were plotted for different time-frequency ranges of interest. In each ERSP plot (Figure 4, Figure 5, Figure 6, Figure 7 and Figure 8), the horizontal and vertical axes represent the time domain and frequency, respectively. The color scale to the right of each ERSP plot displays the non-significant ERSPs (green) and highly significant ERSPs (red), at an FDR-adjusted *p* < 0.05. The delta (1–4 Hz) and theta (4–7 Hz) synchronizations (i.e., increased power in delta–theta band) were detected in the rMFG upon presentation of visual-auditory alarms in SS trials during LHR and RHR inhibitions (see Figure 4). These findings are consistent with previous studies of inhibitory control [34]. Correspondingly, the frontal brain area is actively associated with the response inhibition mechanisms. The delta (1–4 Hz) and theta (4–7 Hz) bands’ power synchronizations, and the alpha (8–12 Hz), beta (13–30 Hz) bands’ power desynchronizations were also observed in the rMFG under SG condition, following the onset of response to the visual stimuli (see Figure 4). The ERSP activities may have been caused by the hand movements in SG condition. The role of frontal cortex in successful inhibitory control was suggested by a significant increase in delta–theta band power, under SS-SG condition, during LHR and RHR inhibitions (see Figure 4).

The ERSP plots in Figure 5 present the delta, theta, alpha, and beta band power synchronizations in the rCG upon presentation of auditory alarms in SS trials under inhibitory control. The delta–theta synchronization was elicited by the visual stimuli in SG trials in LHR and RHR inhibitions. The prominent role of rCG in the response inhibition mechanisms was suggested by significant synchronization of the inhibition related theta-alpha band power, and auditory alarm related delta-beta band powers under SS-SG condition in LHR and RHR inhibitions. The neural signatures, such as theta-alpha band power synchronization under the SS-SG condition in LHR and RHR inhibitions, may be associated with the inhibitory control. 

In addition, Figure 6 shows the auditory-related beta band power synchronization, and theta-alpha band power desynchronization were observed in the rSTG under SS condition in response to the visual-auditory alarms during LHR and RHR inhibitions. The increased power levels in delta, theta, alpha, and beta bands are evident in the rSTG under SG conditions, following the onset of response to the visual stimuli during LHR and RHR inhibitions. The auditory alarm related delta and beta synchronizations were also observed in the rSTG under SS-SG condition during LHR and RHR inhibitions.

In addition, Figure 7 displays the alpha desynchronization following the visual stimuli and auditory alarms in the lSTG under SS-SG condition. Figure 8 reports the synchronization of the delta band power and the desynchronization of alpha band power in the rPL under SS-SG condition during LHR and RHR inhibitions.

Figure 9 displays the power spectral analysis, during visual stimuli and auditory alarms. Pairwise *t*-tests showed significant differences between visual and auditory alarms during human inhibitory control at the rMFG, rCG, and rSTG areas of the brain. We observed that auditory alarms had significantly higher theta and beta band power than the visual stimuli in rMFG (*t* (4) = 3.10, *p* < 0.05; *t* (4) = 2.81, *p* < 0.05; Figure 9). We determined that auditory responses had significantly higher delta, theta, alpha, and beta band power than the visual stimuli’s in rCG (*t* (8) = 1.91, *P* < 0.05; *t* (8) = 2.03, *p* < 0.05; *t* (8) = 2.17, *p* < 0.05; t (8) = 2.23, *p* < 0.05; Figure 9). In addition, we observed that auditory alarms caused significantly higher delta band power than the visual stimuli in rSTG and lSTG (*t* (7) = 2.10, *p* < 0.05; Figure 9). The power change of delta–theta bands is related to the neural mechanism of human inhibitory control in rMFG. In this study, we found inhibition related delta–theta neural makers in rMFG. The changes in EEG activities of the delta and theta bands’ power reveal that rMFG is related to inhibition. These EEG signatures were similar to those obtained in earlier studies of human inhibitory control [34]. Consequently, rMFG is associated with inhibition of the human response. We investigated the prominent role of the rCG in human response inhibition by showing that the inhibition related theta and alpha bands’ power increased significantly. In addition, we observed that delta and beta bands’ power increased during auditory alarms in rCG areas of the brain. However, the rCG has a potential role in executive function and cognitive control, such as in attention, inhibition, and auditory stimulation [32,33]. Furthermore, the power of the delta band increased significantly in both the rSTG and lSTG of the brain, which are related to auditory alarms. The EEG signatures (Figure 9) are related to auditory stimulation in rSTG-lSTG of the brain. Previous research has reported that the superior temporal gyrus (STG) is involved in auditory stimulation [29,30].

#### 3.2.3. The Neural Connectivity of Visual and Auditory Modalities in Human Inhibitory Control

Figure 10 shows the average results of all subjects in visual and auditory modalities. The non-significant connectivity magnitudes are shown in blue and significant connectivity magnitudes (*p* < 0.05) are shown in yellow and red colors [42]. The brain’s neural networks were developed under visual stimuli (RT) during the SG condition in LHR inhibition, as shown in Figure 10. After the presentation of visual stimuli, all the subjects exhibited increased coherence of the delta, theta, alpha, and beta bands’ powers, and higher magnitudes of connectivity between all the neural network pathways, such as the rPL–rCG–rMFG, rMFG–rSTG, rPL–rSTG, and rCG–rSTG (see Figure 10).

#### 3.2.4. Change of Visual to Auditory Neural Networks

The brain’s neural networks were changed under auditory alarm effect (SSRT) during the SS condition in LHR inhibition, as shown in Figure 10. Compared to the visual stimuli (RT), the connectivity magnitudes of neural network, excluding the rCG-rSTG pathway, were significantly higher in the delta, theta, alpha, and beta bands, in response to the auditory effect (SSRT) based on EEG coherence analysis (see Figure 10). Specifically, the neural networks in delta, theta, alpha, and beta bands, and the connectivity magnitudes among the lSTG-rSTG, lSTG-rPL, and lSTG-rMFG neural network pathways, were higher in response to the auditory stimuli those that of the visual stimuli (see Figure 10). In multimodal neural networks between lSTG-rMFG and lSTG-rSTG pathways, higher connectivity magnitudes were observed in theta-alpha and beta bands during auditory modalities compared to the visual modalities (see Figure 10).

Moreover, we also measured neural network pathway response to visual and auditory stimuli during RHR inhibition. We inferred that the human brain neural mechanisms and neural network pathways observed during LHR and RHR inhibitions were relatively similar, as both left and right-hand performed a similar protocol of the response inhibition task. In addition, we investigated how a multimodal neural network was changed under auditory alarms and visual stimuli. Some neural network pathways were highly consistent with those in previous studies of response inhibition [44]. The activation of each brain region was based on the maximum frequency coherence at a specific brain area.

## 4. Discussion

In this study, we investigated how visual stimuli and auditory alarms affect the neural networks in the human brain. As for the behavioral results, the significant differences in the behavioral parameters, such as RT, SSD, and SSRT were identified between visual stimuli and auditory alarms during inhibitory control. The RT observed was significantly greater than the SSD and SSRT in LHR and RHR inhibitions. The RT and SSD values were significantly higher than those obtained previously [45,46]. Significantly higher hit% and lower miss% values were observed for all participants in response to the visual stimuli, in comparison with the response to the auditory alarms. These findings suggested that behaviorally, all participants performed better in response to visual stimuli than auditory alarms. In terms of EEG results, five brain regions, including rMFG, rCG, rPL, rSTG, and lSTG were identified by their similar dipole source locations and power spectra in the brain [35]. The neural mechanisms were investigated in auditory alarms and visual modalities among these five brain regions.

The functional role of the lSTG and rSTG relates to the perception of emotions [29,30]. The lSTG and the rSTG play an important function in auditory stimulation, as well as in source of poor language. It has been reported that the superior temporal gyrus plays an important role in neural network pathways of the prefrontal cortex and the amygdala cortex of the brain; these brain regions have been implicated in the development of social cognition [29,30]. An fMRI analysis study has shown a link between vision-based problem solving and activity in the right anterior superior-temporal gyrus [30]. Moreover, the functional role of rCG is important in inhibitory control, attention control, and motor control. The rCG plays a role in the distribution of cognitive resources in synchronized auditory and visual information [31]. In addition, clinical EEG research has revealed that the rCG has an essential role in neurological disorders, such as schizophrenia and depression [32,33]. The rMFG has been associated with the stages where response inhibition and sustained attention have been proposed to happen [47]. The rMFG is also considered an important brain region for maintaining attention, instead of stopping an action [34,35]. The rPL has been linked to the perception of emotions via facial recognition. The rPL is considered generally related to visual stimuli compared to the human response inhibition [34]. The rPL plays a functional role in the integration of sensory information from various regions of the human brain [36]. The rPL is involved with visuospatial processing in the human brain [37,38].

The main EEG results discussed are the following: The rMFG was identified to have a role in response inhibition, as shown by delta–theta band power synchronizations investigated under the SS-SG condition (see Figure 4). These EEG signatures were similar to those obtained in earlier studies of inhibitory control [34]. Accordingly, the rMFG is associated with response inhibition. The ERSP plot (see Figure 5) revealed the prominent role of the rCG, in response inhibition, by exhibiting significant inhibition related theta and alpha band power synchronizations and auditory alarm related delta and beta band power synchronizations under the SS-SG condition, during LHR and RHR inhibitions [32]. Figure 6 shows the beta band power synchronization and the desynchronizations of the delta-alpha band powers in the rSTG, during inhibitory control in the SS-SG condition. These EEG signatures are related to the auditory alarm response [48].

Cross-modal interference occurs when incompatible information is delivered by different senses. For example, when we listen to someone’s voice, watching the movement of their lips can improve speech intelligibility under noisy conditions [49,50,51]. When the EEG signals of visual stimuli and auditory alarms do not coincide, as for a person who is seeing another person speaking one syllable of speech while listening to another, the listener characteristically reports hearing a third syllable characterizing a combination of what he saw and heard [52]. Revealing where and how multisensory information is combined and processed in the human brain is fundamental to understanding how auditory inputs affect auditory perception and behavior [53]. Additionally, various brain regions, including primary areas, sometimes receive different inputs from more than one of the senses, as occurs in the auditory cortex, which in humans is sensitive to some visual stimuli, as well [54,55]. Visual stimuli and auditory alarms can modulate neural oscillations in the primary auditory cortex [56]; a facilitator role for the above inputs in sound localization has been suggested [57], and the finding supports that visual inputs can increase the amount of spatial information that is transferred by neurons in the auditory cortex [58]. In addition, visual stimulation is important to guide the development of the auditory spatial response properties of neurons in certain regions of the brain. This is most evident in the superior colliculus of the midbrain, where perceptible visual and auditory inputs are organized into topographically associated spatial maps [59]. This organization allows each of the sensory inputs that are associated with a particular event to be converted into suitable motor signals that alter the direction of a gaze. However, individual neurons of the superior colliculus receive convergent multisensory inputs, and the strongest responses are often generated by stimuli in close temporal and spatial proximity [60].

Visual and auditory modalities commonly work together to facilitate the identification and localization of objects and events in the real world. The human brain critically establishes and maintains consistent visual and auditory stimulation. Therefore, different sensory signals that are associated with different objects that can be seen and heard can still be correctly synthesized together [11]. Considering the impacts of visual and auditory stimuli in multisensory integration, where cross-model response inhibition causes stronger responses than auditory alarms or visual stimuli alone, only the power of the theta band was found to increase [61]. This outcome is consistent with other studies that have identified the effects of multisensory integration within the theta band [62]. The spatial distribution of this response (i.e., theta band power) is consistent with formerly determined topographies that are present in frontal-central, as well as occipital, lobes of the brain [63]. The delta and theta bands’ powers reveal interesting effects of audiovisual relationships [64]. In this work, delta–theta band power was observed in posterior brain regions in response to visual and auditory stimuli, suggesting visual and auditory stimuli-related EEG signatures in the rSTG and rPL of the brain [48]. The EEG signatures in the auditory cortex have higher neural activations than those in the visual cortex. Unisensory auditory alarms can evoke responses in the sensory cortex that have not been specified for visual sensory domain [65]. This work suggested the inhibition related theta and beta band power synchronization, and the auditory alarm related delta and beta band power synchronization, occurred in the rCG during inhibitory control. Therefore, the rCG has a role in executive function and cognitive control, which involves a set of cognitive processes, which include inhibitory control, attention control, and cognitive processes. Accordingly, clinical research has revealed that the rCG has an essential role in neurological disorders, such as schizophrenia and depression [32,33]. In this study, inhibitory control findings concerning the theta band power in the frontal cortex was identified as an EEG signature related to those identified in previous studies [34].

In this work, we investigated how auditory alarms affect neural network pathways during visual stimuli. In this investigation, the first time, we found that the theta-alpha and beta bands between lSTG–rMFG and lSTG–rSTG pathways had higher connectivity strength in the brain’s network when performing the visual tasks changed to receiving the auditory alarms. Previous studies reported that the functional connectivity role of lSTG and the rSTG have been involved in the perception of emotions [29,30]. The lSTG–rSTG plays an important role in auditory stimulation, as well as in language development. Preceding studies have demonstrated that lSTG–rSTG played a significant functional role in the pathway between the prefrontal and the amygdala cortices [29,30]. Former clinical research reported that patients with schizophrenia have been found structural-functional abnormalities in their lSTG–rSTG [29,30]. Furthermore, the significantly increased power of the delta band has been related to auditory stimulation in rSTG and lSTG of the brain. These neural signatures have been linked to the auditory stimulation in rSTG and lSTG of the brain [29,30]. The rMFG has been connected to human inhibition and visual attention. The rMFG has played a functional role in controlling visual attention. Consequently, rMFG has been considered as an important brain region to investigate visual attention instead of inhibitory control [34,35]. The increased power of delta and theta bands has been related to the neural activities of human response inhibition in rMFG. In our study, we observed delta and theta bands related to inhibition (i.e., neural signatures) in the rMFG of the brain. These neural signatures discovered that rMFG has an important functional role in human response inhibition. These neural signatures were observed similarly to those investigated in previous studies of response inhibition [34]. Moreover, the functional connectivity function of rCG has been observed during motor control, human inhibitory control, and visual attention. The rCG has generated neural impulses that control the movement of the hand. The rCG played a role in the neural pathways of auditory and visual stimulations [31]. Accordingly, a clinical study reported that the rCG plays an essential role in schizophrenia and depression [32,33]. We observed the functional role of the rCG in human inhibition. Then, we found that the theta and alpha band powers increased significantly during response inhibition in the rCG of the brain. In addition, we investigated that delta and beta band powers increased during auditory alarms in the rCG of the brain. Moreover, the rPL has been related to visual stimulation [34]. The rPL revealed a functional role in the integration of sensory information from various regions of human brain [36]. The functional role of the rPL has been found in visuospatial processing in the human brain [37]. Finally, novel neural network pathways were investigated between the lSTG, rSTG, rCG, rMFG and rPL of the brain during visual and auditory stimulation.

Further, such higher network connectivity patterns were more pronounced in SS trials than those of SG trials. These findings suggest that different users have higher levels of connectivity in the neural network spanning various brain regions for auditory alarm (i.e., inhibitory control). The EEG results implied that all study participants exhibited higher inhibition-induced neural oscillations in the rCG from auditory alarms than from visual stimuli. In summary, the novel visual to auditory neural network pathways under human inhibitory control were developed from the EEG based analysis, and some pathways of this brain network are consistent with previous response inhibition studies [44]. As a final point, the limitations of the present study are worth stating. All subjects were males, which may lead to difficulty in generalizing the EEG results to both genders. The experimental scenario in our work was adapted from a well-known stop signal task, which uses only 2D stimuli and may not be as representative as more realistic environments. Future work may construct a real-world scenario with 3D-virtual reality and augmented reality to improve the real-world compatibility of this study. In the absence of such advanced technologies, we can assume that the brain network model of visual and auditory perception is accurate to normal conditions and attempt to experimentally strengthen the principles described in this study. However, more research is required for an entire explanation of the emerging neural network model, in both physiological and cognitive terms. In future studies, we will increase the number of participants and perform inter- and intra-subject analyses.

## 5. Conclusions

To conclude, in this work we use advanced EEG-dipole source localization methods with independent component analysis (ICA) and EEG-coherence analysis, to investigate the effect of auditory alarms on visual stimuli in the human brain. In this study, we investigated the novel brain network pathways between five activated brain regions that included: lSTG, rSTG, rCG, rMFG, and rPL, under the effects of visual stimuli and auditory alarms. The present study revealed how visual stimuli and auditory alarms under inhibitory control affect the neural mechanisms in the brain. A visual-auditory-cross brain network model was developed to demonstrate the associations among the five activated brain regions, in the form of different neural pathways related to the visual and auditory modalities under response inhibition. We inferred that the connectivity magnitudes of the neural network in theta-alpha band powers were found to be higher in response to auditory alarms compared to visual stimuli. In addition, the EEG results implied that all study participants exhibited higher inhibition-induced neural oscillations in the rCG under auditory alarms compared to visual stimuli. These results provided new knowledge about changes in neural network pathways from visual stimuli to auditory alarms in a realistic environment.

## Figures and Tables

**Figure 1 brainsci-09-00216-f001:**
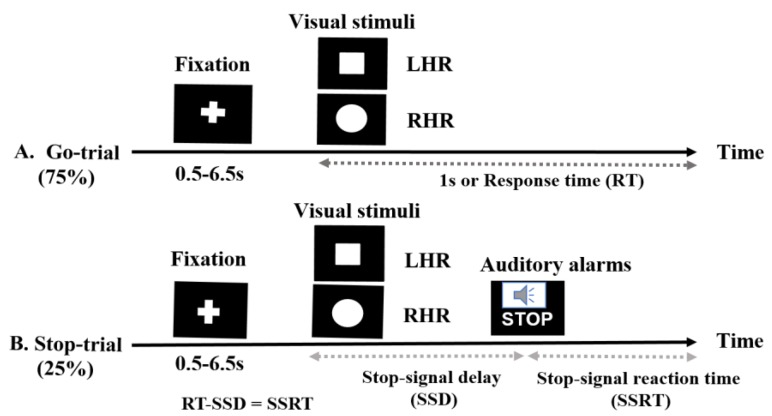
The stop signal task used for visual stimuli and auditory alarms: (**A**) In the go-trials, participants responded to the shape of a go stimulus (a “square” requires a left-hand response (LHR) and a “circle” requires a right-hand response (RHR). The square and a circle shapes were used as visual stimuli. (**B**) In the stop-trials, a beep sound (auditory alarm) was used as a stop signal, to instruct the participants to control their response. The behavioral parameters measured in this experiment included fixation, reaction time (RT), stop signal delay (SSD) and stop signal reaction time (SSRT).

**Figure 2 brainsci-09-00216-f002:**
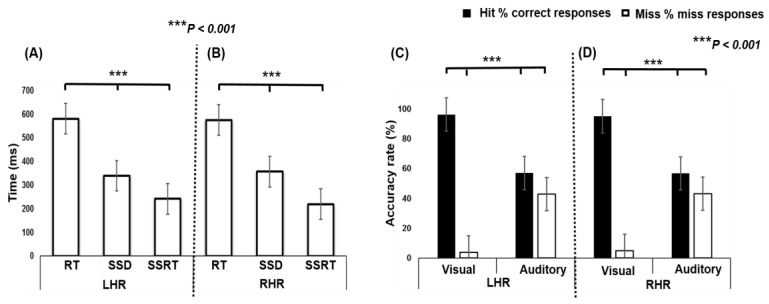
(**A**) Behavioral outcomes between visual stimuli and auditory alarms during LHR inhibition. (**B**) Behavioral results of visual stimuli and auditory alarms in RHR inhibition. Asterisks indicate pairwise significance difference (*** *p* < 0.001) in ANOVA: Single factor between the RT, SSD, and SSRT conditions. (**C**,**D**) Comparisons between hit% (i.e., accuracy rate) and miss% (i.e., inaccuracy rate) while responding to the visual-auditory stimuli during LHR and RHR inhibitions. Asterisks show significance difference in ANOVA between the hit% and miss% with visual stimuli and auditory alarms.

**Figure 3 brainsci-09-00216-f003:**
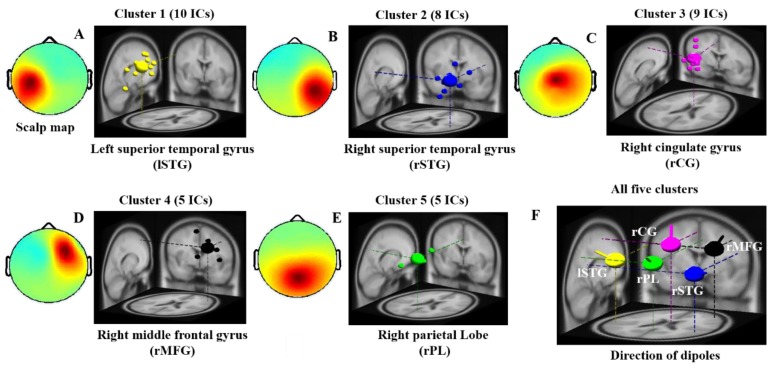
Scalp maps and dipole source locations of five groups of independent components (IC) in all subjects. Right panel: Plot of 3D dipole source locations and their projections onto the MNI brain template: (**A**) left superior temporal gyrus (lSTG), (**B**) right superior temporal gyrus (rSTG), (**C**) right cingulate gyrus (rCG), (**D**) right middle front gyrus (rMFG), (**E**) right parietal lobe (rPL) and (**F**) the direction of five dipoles. Left panel: average maps of the scalp from all independent components within a cluster. Cluster 1—left-superior temporal gyrus (*n* = 10). Cluster 2—right-superior temporal gyrus (*n* = 8). Cluster 3—right-cingulate gyrus (*n* = 9). Cluster 4—right-middle frontal gyrus (*n* = 5). Cluster 5—right-parietal lobe (*n* = 5). *n* is the number of diploes estimated in a cluster.

**Figure 4 brainsci-09-00216-f004:**
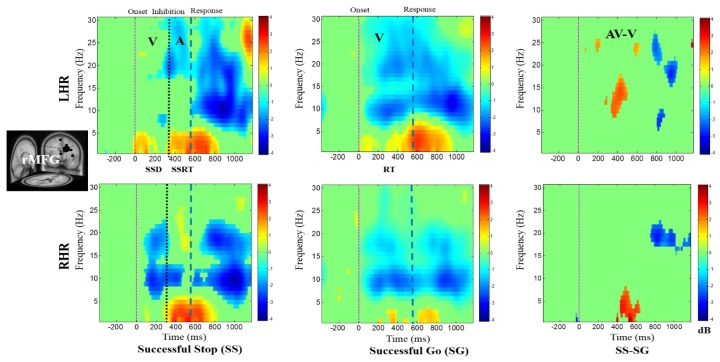
The event-related spectral perturbation (ERSP) plots showing the post-stimuli EEG modulations in right middle frontal gyrus (rMFG) of the brain under combined visual (V) stimuli and auditory (A) alarms. The successful go (SG) trial is elicited by only visual stimuli, and the successful stop (SS) trial is elicited by auditory alarms. In (SS-SG) condition, the ERSP plots display the EEG modulation of auditory alarms by comparing the ERSP plots of audio-visual (AV) and visual V) stimuli; (AV-V). Purple dashed line: Onset of the visual stimuli. Black dashed line: Onset of the auditory alarms. Blue dashed line: Onset of response. Color bars show the amplitude of the ERSP; statistical threshold at *p* < 0.01.

**Figure 5 brainsci-09-00216-f005:**
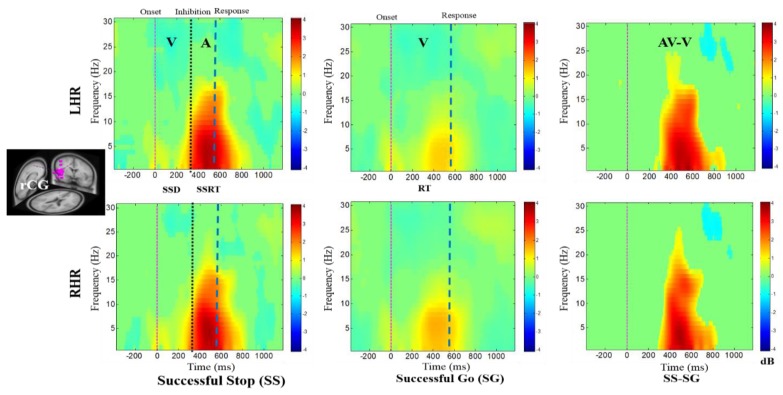
The ERSP plots showing the post-stimuli EEG modulations in the right cingulate gyrus (rCG) of the brain under combined visual (V) stimuli and auditory (A) alarms. The successful go (SG) trial is elicited by only visual stimuli, and the successful stop (SS) trial is elicited by auditory alarms. In (SS-SG) condition, the ERSP plots display the EEG modulation of auditory alarms by comparing the ERSP plots of audio-visual (AV) and visual (V) stimuli; (AV-V). Purple dashed line: Onset of the visual stimuli; black dashed line: Onset of the auditory alarms; blue dashed line: Onset of response. Color bars show the amplitude of the ERSP; statistical threshold at *p* < 0.01.

**Figure 6 brainsci-09-00216-f006:**
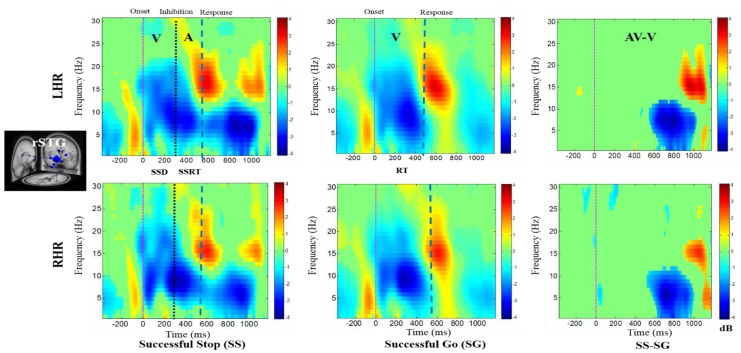
The ERSP plots showing the post-stimuli modulations in right superior temporal gyrus (rSTG) of the brain under combined visual (V) stimuli and auditory (A) alarms. Purple dashed line: Onset of the visual stimuli; black dashed line: Onset of the auditory alarms; blue dashed line: Onset of response. Color bars show the amplitude of the ERSP; statistical threshold at *p* < 0.01.

**Figure 7 brainsci-09-00216-f007:**
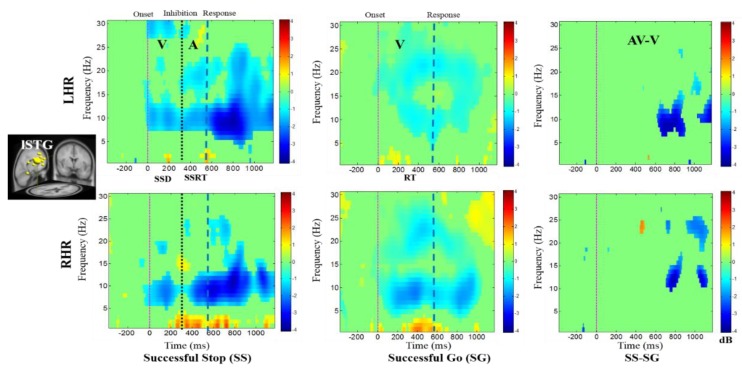
The ERSP plots showing the post-stimuli modulations in left superior temporal gyrus (lSTG) of the brain under combined visual (V) stimuli and auditory (A) alarms. Purple dashed line: Onset of the visual stimuli; black dashed line: Onset of the auditory alarms; blue dashed line: Onset of response. Color bars show the amplitude of the ERSP; statistical threshold at *p* < 0.01.

**Figure 8 brainsci-09-00216-f008:**
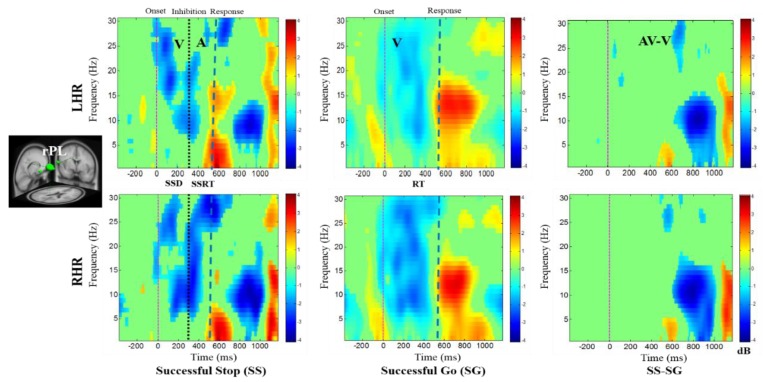
The ERSP plots showing the post-stimuli modulations in right parietal lobe (rPL) of the brain under combined visual (V) stimuli and auditory (A) alarms. Purple dashed line: Onset of the visual stimuli; black dashed line: Onset of the auditory alarms; blue dashed line: Onset of response. Color bars show the amplitude of the ERSP; statistical threshold at *p* < 0.01.

**Figure 9 brainsci-09-00216-f009:**
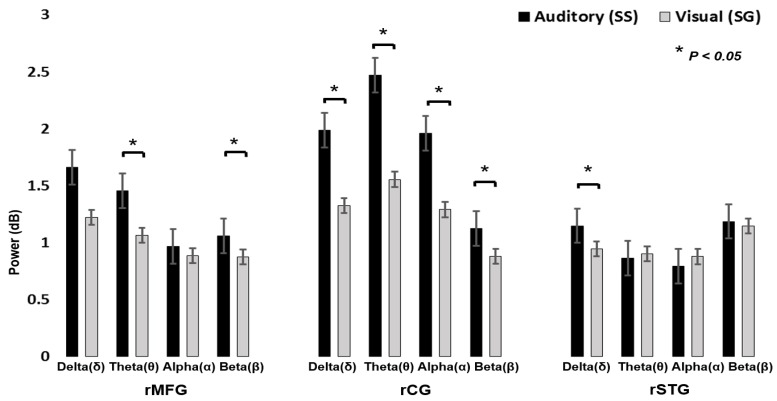
The comparison in absolute value of delta (δ), theta (θ), alpha (α), and beta (β) band powers and standard errors between visual stimuli and auditory alarms in rMFG, rCG, and rSTG during human inhibitory control. Asterisks show pairwise significance (* *p* < 0.05) in t-tests between the visual and auditory conditions.

**Figure 10 brainsci-09-00216-f010:**
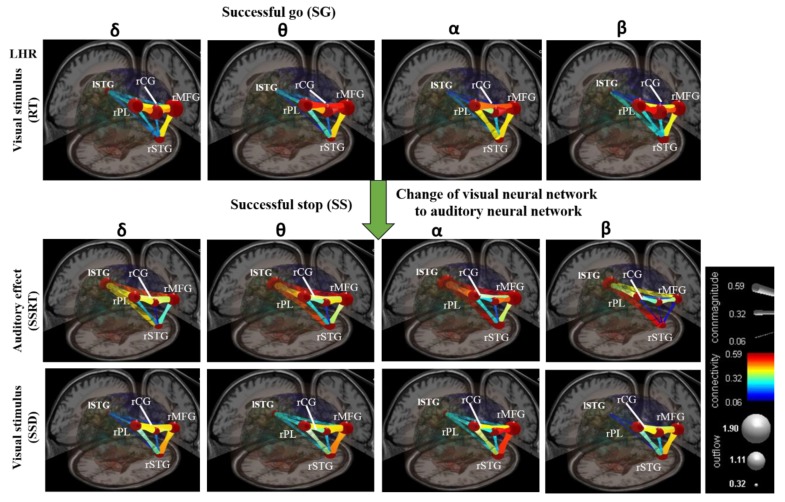
The design of visual and auditory cross-model neural networks under human inhibitory control of left-hand response (LHR). The neural network between five brain regions, including left superior temporal gyrus (lSTG), right superior temporal gyrus (rSTG), right cingulate gyrus (rCG), right middle front gyrus (rMFG), and right parietal lobe (rPL). The green arrow shows the change of the visual neural network to the auditory neural network. Color bars shows the scale of the connectivity strength; statistical threshold at *p* < 0.01. The outflow was obtained between two dipole sources. The gray node shows the high and low outflow strength between the two dipoles. Connectivity (edge color mapping): The color of the edges was mapped to connectivity strength (amount of information flow along that edge). Red = high connectivity and green = low connectivity. ConnMagnitude (edge size mapping): The size of edges of the graph (connecting “arrows”) was mapped to connectivity magnitude (i.e., absolute value of connectivity strength).

**Table 1 brainsci-09-00216-t001:** Five dipole clusters in human brain, and the Montreal Neurological Institute (MNI) coordinates of their source distributions during visual and auditory stimuli, under human response inhibition.

Component Clusters	Side	Brain Regions	MNI Coordinates (mm)			Cluster Size (voxels)
			X	Y	Z	
1	Left	Superior Temporal Gyrus	−59	−25	30	18
2	Right	Superior Temporal Gyrus	51	−40	24	12
3	Right	Cingulate Gyrus	0	4	56	15
4	Right	Middle Frontal Gyrus	43	36	45	59
5	Right	Parietal Lobe	5	−60	42	55

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
