# Peer review of "Modulation of the Visual to Auditory Human Inhibitory Brain Network: An EEG Dipole Source Localization Study"

_brainsci, 2019, doi:10.3390/brainsci9090216_

Round 1
Reviewer 1 Report
This paper investigates the behavioral and EEG aspects derived by a task combining visual and auditory stimuli. The authors observed increased power in certain frequency bands of EEG in several regions of the brain cortex such as right middle frontal gyrus, right cingulate gyrus and left superior temporal gyrus, which may be associated with the neural networks participating in this visual-auditory task.
This is an interesting study that could lead to further understanding of the neural network regarding the cognitive function of the human brain. Following points may be valuable to be addressed.
1) The authors specifically focus on the five regions of the cortex, left and right superior temporal, right cingulate, right middle frontal gyri and right parietal lobe. It seems unclear whether these regions of interest (ROIs) are deta-driven or obtained in an arbitrary manner.
2) In each ROIs, the significant difference of the power is seen in different bands as shown in Fig. 7. It would be informative to describe the possible role of these bands in the cognitive function.
Author Response
Dear Reviewer,
Please see the attached file of responses. Many thanks.

Reviewer 2 Report
Review Brain Sciences
Manuscript: brainsci_552561
Title: „Modulation of visual to auditory human inhibitory brain network: an EEG dipole source localisation study“
Authors: Rupesh Kumar Chikara & Li-Wei Ko
Summary: The manuscript „Modulation of visual to auditory human inhibitory brain network: an EEG dipole source localisation study“ describes a study using a newly designed stop signal task to examine spatio-temporal characteristics of EEG source space data on auditory triggered inhibitory control behaviour. The authors applied different sound methodological approaches, such as time frequency analyses, source localisation, and coherency analyses to examine the data of 12 study participants. Main result and take home message was that there was no main result in the classic sense, but a valuable bunch of exploratory insights into the dynamic neural processing of some of the participating brain areas in a visual signal choice task that was modulated by auditory signal induced inhibition processing. The study provides a substancial contribution to the topic by using advanced methodological approaches. Some points should, however, be considered before publication – see comments below.
Major comments:
-> One of the main critical issues is the small sample size of 12 participants. On the other hand, the authors present very sophisticated and detailed analyses of their data, which might counterbalance the lack of an adequate sample size to some degree. At first glance, an experienced reader might assume that here are data presented from a PhD or master thesis that shall be published fastly or as a by-product of a bigger project. If this would not be the case, the authors would perhaps at least double the number of study participants and additionally check for robustness and reliability of their data via intra-session split-half procedures or others. Here, the action editor has to finally decide.
-> Important: Auditory alarms in daily life are rather associated to stereotyped action sequences individually learned during lifelong learning histories (i.e., associated in perception-action-cycles; see Fuster, 2006, 2009; Fehr et al., 2014; etc.). Therefore, the authors should be careful when they extrapolate or generalize their results, due to an arbitrary experimental context, to real life problems outside the laboratory. Limitations should be formulated accordingly.
-> Please consider that auditory alarm trials were presented substantially less frequent than pure visual choice trials; this qualifies the the experiment to an oddball design. Thus, no only the experimental condition (visual choice versus inhibition events) but also the frequency of conditional scenarios (inhibition event less frequently presented) contribute to the modulation of neural correlates. Frequency- and condition-effects cannot be disentangled at all, and this has to be carefully discussed as a critical limitation of the study design. In future studies, the authors should, if they insist in using oddball designs, provide at least two less frequent task conditions of interest to be contrasted to each other.
-> Please check the whole manuscript for language related flaws and readability.
Detailed comments:
Abstract:
-> Authors should introduce their experimental design with at least a phrase in the abstract that the reader can handle expression like “… the visual stimulus elicited …”. Abstract is a stand-alone text that should be sufficiently self-explaining.
-> In the title, the authors refer to dipole analyses, then, they continue with behavioral data in the abstract, followed by reporting ICA and coherency analyses as methods, and closing with reporting regional frequency data. I think, the prelude of the manuscript would tremendously benefit from a clearer line and focussing on the bullet points of the work.
-> Page 1, lines 26 and 27: “… findings could be helpful …” Why and in what way? Please concretize and give comprehensible examples.
Introduction:
-> One might speculate whether each issue mentioned in the introduction section is really necessary. On the other side, I think it is helpful to shortly describe or even explain basic principles of the advanced methods used in the study to consider also those readers who are not that educated in the respective methodological domains.
-> I understand the exploratory nature of the study, however, the introduction section should at least result in some close working hypotheses based on the big bunch of previously cited literature in the section. These hypotheses should, beside all the very interesting exploratory and consecutively hypotheses generating results, adequately tested and their verifications or falsifications should be respectively discussed in the discussion section. From my point of view, this is mandatory to get a closeer link to previous work on similar topics and it would enhance the value and scientific integrativness of the present work.
Methods:
-> As I hopefully understood right, the authors applied an oddball design (see also major comments above). In the present case, inhibition processing might be strongly biased or modulated by the processing of less frequent events (i.e., frequency of events as a discrete variable). Thus, with this design, it is not possible to draw a clear line between inhibition processing and the common processing of less frequent events and its neural correlates. This is a major critical point that has to be clearly stated as a limitation of the present study.
-> It appears that the stepwise adaptation of the SSD to “optimize” inhibition behaviour to chance level (50 % performance in a binary event space: inhibit versus non-inhibit) makes any kind of sense. However, there are 48 stop trials in all. Over how many trials was a stable 50 percent performance detected, and how many trials were then left to be analyzed (minus those that were artefact-contaminated)? Detailed information about this is necessary.
-> Page 4, lines 153-154: Please exactly describe how you calculated the critical SSD, and what was exactly subtracted as RT? The Mean RT? RTs from which period of the run? Across the whole run or just from that time point, where the critical SSD was reached to the end? Please specify.
-> Page 5, line 203: Why do the authors apply Bonferroni correction for dependent statistical cells? That´s wrong, as Bonferroni-calculus affords statistical independency. The respective mal-usage of Bonferroni-correction on dependent statistical cells usually produce an inacceptable level of beta-error – please consider this (Liebermann & Cunningham, 2009).
Results:
-> Section 3.1.: Why have the authors differenciated between significance levels .05, .01, and .001? Significant is significant – enough said.
-> Just an annotation to section 3.2.: I would never trust an ICA-procedure for sufficient artefact detection alone. A visual trial-by-trial raw data inspection appears to be indispensable for all biosignal analyses such as EEG, MEG, and so forth.
-> Section 3.2.1.: The way, how dipole locations were determined/fitted has to be explained much more comprehensible as it is one of the main results of the present work. How many data trials were included per condition? What were the fitting criteria? What means 50 to 70 % of the subjects/participants – 6 to 8.4? According to which criterium were the scalp maps accepted as similar? Please convince the reader that the procedure have not followed just arbitrary criteria, and please try to arrange this section much clearer and comprehensible for the reader – make it understandable, - I think it’s worthwhile.
-> Figure 7: Can you please display standard deviations instead of standard errors – or what is it?
-> Power Spectral Analyses: Have you balanced the number of trials included in individual and condition-related (i.e., visual and auditory) analyses? What is the reason for this analysis?
Discussion and conclusion:
-> Page 11, lines 382-384: “These findings suggested that all participants performed better in response to the visual than in response to the auditory alarms.” What is the visual alarm? I hope that I understood it right when the visual stimulus provides a cue for either RH or LH GO-response and the auditory stimulus transforms the GO-scenario into a NoGo-scenario, which is qualitatively seen something completey different. Strictly thought is the direct comparison between both behavioural and neural correlates of visual detection triggered response behaviour and auditory detection triggered inhibitory non-response behaviour a bit like comparing apples with oranges. It appears that this is the reason why the authors seem to have difficulties to find a handy explanation or ‘take home message’ based on their results. Without a convincing conclusion based on comprehensible hypotheses, the present data can only be seen as an exploration of neural correlates that were examined because the authors were able to examine them – to say, just arbitrary motivated.
-> Furthermore, any potential conclusion based on the presented data have to be seen in relation to the reproducability of the results. Thus, not only validity, but also reliability has to be discussed. The authors should provide concrete ideas and plans for following up their work in consecutive studies to ensure that the reader does not get the impression that their results are just a mayfly.
-> Ideas about how the present concept might be proved by more realistic stimulation procedures: Additional variables and the consideration of individual parameters might help to shed more light into the neural processing of realistic context (cmp. Fehr & Milz, 2019; Fehr, 2013). Data are ok, and the methods applied seem sophisticated, however, what is the concrete benefit of the results? More real-life-related visions and impulses for consecutive science on the topic are needed.
… Page 13, lines 449 and 450: “users”? You meant study participants, I guess. I would also consequently use the term “study participants” or individuals “instead” of “subjects”.
… Page 13, lines 455-456: “All subjects were males, which may lead to difficulty in simplifying the EEG results to both genders.” In small samples this is called “sample homogenization” an therefore completely ok to reduce noise potentially produced by heterogenity in those small groups. In bigger samples with more than 28 cases per considered dimension (e.g., age-cohort, gender group, patients and controls, and so forth) balancing of sub-groups might make sense, but surely not in a mini-sample of 12 individuals that is anyhow not extrapolatable to population; particularly without any reliability proofs.
References:
Fehr T. (2013). A hybrid model for the neural representation of complex mental processing in the human brain. Cognitive Neurodynamics 2013; 7: 89-103.
Fehr T & Milz P (2019). The individuality index: a measure to quantify the degree of interindividual, spatial variability in intra-cerebral brain electric and metabolic activity. Cognitive Neurodynamics, DOI: 10.1007/s11571-019-09538-9.
Fehr T, Achtziger A, Roth G, Strüber D. (2014). Neural correlates of the empathic perceptual processing of realistic social interaction scenarios displayed from a first-order perspective. Brain Research, 1583, 141-158.
Fuster JM (2006). The cognit: A network model of cortical representation. Int J Psychophysiol, 60, 125-132.
Fuster JM (2009). Cortex and memory: Emergence of a new paradigm. Journal of Cognitive Neuroscience, 21, 2047-2072.
Lieberman MD, Cunningham WA. (2009). Type I and Type II error concerns in fMRI research: re-balancing the scale. Social, Cognitive, and Affective Neuroscience, 4, 423-428.
Author Response
Dear Reviewer,
Please see the attached file of our responses. Thank you very much.

Reviewer 3 Report
The authors studied neural mechanisms underlying response inhibition in the stop-signal paradigm. They measured EEG and behavioral performance in this task and analyzed these data using ICA, coherence, and dipole methods.
I have some major concerns:
First, the paper is not well written in terms of expression and logic flow probably due to very poor English writing skills. It should be greatly improved - not only typos and grammars, but also transition between sentences and between paragraphs.
Second, the tasks are not well designed to investigate the response inhibition. Specifically, the authors simply considered success and miss trials during the stop-signal conditions. However, they did not describe how they handled the trials when participants moved but successfully stopped. In other words, the current settings cannot perfectly isolate auditory perception as well as (inhibitory) motor control from the response inhibition without monitoring EMG from participants. Alternatively, they may introduce one more condition as a baseline; for example, an observation of stop-signal condition, in which the participants can simply watch the current stop-signal condition without motor response. Anyway, the key point is that the authors have to provide neural mechanisms underlying isolated response inhibition.
Third, by employing a data-driven approach, the authors determined five brain regions as critical areas for visual and auditory processing. They must validate the functional roles of these regions in Introduction with literature review.
Some minor comments (basically lack of details):
Line 111, biased results due to the handedness
Lines 119-121 unclear.
Lines 125-126, how did the participants respond to the stimulus?
Figure 1, 1-SSD?
Line 139, what is the size of visual stimuli?
Line 148, SSD and 250ms are unclear.
Line 150-151, when do you use +50ms and -50ms?
Line 153-154, citation for your assumption.
Line 154, 48/192 is not 25%.
Line 160, which artifacts rejection methods and why the size is reduced?
Line 161, which IIR filter and settings (not asking 1 and 50 Hz) were used?
Line 163-165, which criteria were used to identify artifacts?
Line 167-168, which ICA algorithm was used?
Line 169, missing 0ms information for -500-1300ms epochs.
Line 171-, This section is not for behavior but for EEG. Did you
Line 183, channels or components for ERSP? Which options for ERSP?
Line 187-, coherence cannot reveal brain activity without volume conduction.
Line 195-197, clustering method and brain regions. More details for the “variances among their visual and auditory modalities.”
Line 199-, you should show normality test and equality of variances for the pairwise t-test. The t-test is not enough to reveal the interaction between your variables. Your “Post hoc comparisons” is in an ANOVA context. As such, you should use ANOVA and post hoc comparisons.
Line 205-207, visual stimuli exist both in go and stop trials. How did you use visual stimuli? In the paper, what are the differences between “go and stop trials” and “visual and auditory stimuli,” because it seems that you mix-used them. Use only one unless you have clear difference.
Line 207-209, unclear, mean ERSP was computed per cluster?
Line 209, how did you compute the power spectrums? Provide the steps and all parameters.
Line 210, define the frequency ranges for your EEG bands.
Line 211, which statistical approach was used for coherence?
Figure 2, specify the number of asterisks and the significance levels.
Line 245-, most of the texts is not the result but the method. Some of them are discussion.
Line 248, how many dipoles were estimated per clusters? What are the parameters for DIPFIT2?
Line 250, k-means of which values?
Line 255, “less than 50-70% of subjects have it” is unclear.
Line 257, how did you find the Talairach coordinates for your clusters? Maximum activity?
Line 260-262, “directed the analysis of the temporal resolution of EEG signals” unclear.
Figure 3 and Table 1, use a common coordinate. MNI is the internal standard now, so convert Talairach to MNI. Figure 3 does not show the direction of dipoles.
Line 270-, most of the texts are not results. Figure 7 does not show ERSP.
Figure 4-6, color bar does not represent “power magnitude of the ERSP,” but represent the relative spectral log amplitude, or simply the amplitude scales.
Page 10 and the supplementary materials, what is the major reason of providing the other two regions in the supplementary document? Why don’t you move all supplementary results to the results and add texts?
Figure 8, what does the green arrow mean? The small legend is hard to read.
Line 385, what do you mean “power spectral profiles?” ERSP?
Author Response
Dear Reviewer,
Please see the attached file of our responses. Many thanks.

Round 2
Reviewer 2 Report
The authors made a great job in revising the original manuscript and considered all important comments.
Author Response
Dear Reviewer,
Please find the attached file of our response. Thank you very much for providing valuable comments and suggestions.

Reviewer 3 Report
The manuscript is greatly improved including the goal. Particularly, the goal is clearly written to study the neural changes on visual attentive processing due to unexpected auditory events. However, it still contains lots of English-related problems.
I have some minor comments.
The purpose of your IIR filter is not limited to the de-trending of the signals, since it is a band-pass filter. Add IIR filter type (e.g., Butterworth, etc.) and related information (e.g., filter order, etc.).
You specified three measures (scalp map, power spectrum, and dipole location) that were used for artifact identification, but did not provide the criteria for each of them. You might execute this step by visual inspection if you didn’t use any artifact identification algorithm.
You might use the InfoMax ICA if you used runica.
You might use only one head model (i.e., either a spherical model or a BEM model) in DIPFIT2.
In your response #31, the k-means used the coordinates of dipoles, but in your texts, you wrote “component clustering” or “IC clusters”. Did you use the k-means both for components and for dipoles? Make it clear in the entire manuscript. In addition, provide clear descriptions about the k-means step, particularly, what was the input for the k-means of IC clusters?
Double check the coordinates in Table I. I can see a track change of MNI from Talairach, but no track changes in the coordinate values.
In Figure 10, explain the other two legends (outflows and connmagnitude).
Finally, in discussion, provide more deep insights for the connectivity results (i.e., Figure 10), which include connectivity in specific regions and in specific frequency bands in the context of the tasks.
Author Response
Dear Reviewer,
Please find the attached file of our responses. Thank you very much for giving valuable suggestions and comments.
